# Spatial distribution of clinical computer systems in primary care in England in 2016 and implications for primary care electronic medical record databases: a cross-sectional population study

Evangelos Kontopantelis,[1,2] Richard John Stevens,[3] Peter J Helms,[4] Duncan Edwards,[5] Tim Doran,[6] Darren M Ashcroft[1,7]

For numbered affiliations see end of article.

**Correspondence to**
Professor Evangelos Kontopantelis;
e.kontopantelis@manchester.ac.uk

## ABSTRACT

**Objectives** UK primary care databases (PCDs) are used by researchers worldwide to inform clinical practice. These databases have been primarily tied to single clinical computer systems, but little is known about the adoption of these systems by primary care practices or their geographical representativeness. We explore the spatial distribution of clinical computing systems and discuss the implications for the longevity and regional representativeness of these resources.

**Design** Cross-sectional study.

**Setting** English primary care clinical computer systems.

**Participants** 7526 general practices in August 2016.

**Methods** Spatial mapping of family practices in England in 2016 by clinical computer system at two geographical levels, the lower Clinical Commissioning Group (CCG, 209 units) and the higher National Health Service regions (14 units). Data for practices included numbers of doctors, nurses and patients, and area deprivation.

**Results** Of 7526 practices, Egton Medical Information Systems (EMIS) was used in 4199 (56%), SystmOne in 2552 (34%) and Vision in 636 (9%). Great regional variability was observed for all systems, with EMIS having a stronger presence in the West of England, London and the South; SystmOne in the East and some regions in the South; and Vision in London, the South, Greater Manchester and Birmingham.

**Conclusions** PCDs based on single clinical computer systems are geographically clustered in England. For example, Clinical Practice Research Datalink and The Health Improvement Network, the most popular primary care databases in terms of research outputs, are based on the Vision clinical computer system, used by <10% of practices and heavily concentrated in three major conurbations and the South. Researchers need to be aware of the analytical challenges posed by clustering, and barriers to accessing alternative PCDs need to be removed.

## Strengths and limitations of this study

► Cross-sectional analysis of all clinical computer systems used in English primary care, in August 2016.

► Data allowed a detailed description of regional use of each clinical computer system at the Clinical Commissioning Group level, and the discussion of implications for UK primary care databases (PCDs).

► Although regional presence of a clinical computer system cannot be equated to contribution to a PCD, since contributing practices are anonymised, inferences on the regional representativeness of UK PCDs are still possible.

purchase of clinical computer systems in 1998 and full subsidies in 2003 (in anticipation of the implementation of a national pay-for-performance programme) UK primary care became fully computerised.[2 3] Interoperability requirements of the National Health Service (NHS) led to the universal adoption of a loosely hierarchical clinical coding system, known as Read codes,[4] which is due to be replaced in April 2018 by a multihierarchical coding system (SNOMED CT).[5] Various commercial providers were permitted to enter the market for clinical computer systems, resulting in numerous different systems with varying interfaces, mechanisms and implementations of Read code usage.[6] By 2010–2011, seven clinical computer systems were consistently active in England, holding 99% of the market share: EMIS systems (LV and PCS) were active in 54.7% of practices, followed by Vision V.3 (18.1%) and SystmOne (17.8%), with the remaining 9.4% held by other systems (Synergy, Practice Manager, Premiere and the then newly launched EMIS Web).[2]

## INTRODUCTION

Primary care in the UK has been almost fully computerised since the early 1990s.[1] Following the introduction of partial subsidies for the

The uniformity and interoperability standards have facilitated the creation of large repositories of primary care electronic health records (EHRs), which contain the complete primary care records of patients attending general practices in the UK. The secondary use of these EHRs by researchers, both within and outside the UK, has been increasing exponentially,[7] and they have provided insights in numerous research areas, including real-world effectiveness, adverse events, resource utilisation, condition prevalence and incidence, quality of care and policy interventions.[8 9] Several EHR databases exist, maintained by the different clinical computer system providers, drawing data from practices using their systems that have agreed to make patient data available for secondary use. The four largest EHR databases (hereafter primary care databases or PCDs) in terms of numbers of patient records are the Clinical Practice Research Datalink (CPRD), The Health Improvement Network (THIN), QResearch and ResearchOne.

The CPRD (formally General Practice Research Database) was established in 1987 and has been owned by the Secretary of State for Health since 1994. In May 2017, the CPRD covered approximately 8% of the UK population, with 718 contributing general practices and over 17 million total patients (historical and current). The CPRD primarily collects data from Vision practices, although it is currently undergoing an expansion to include EMIS practices, and a future expansion to cover SystmOne practices is planned. THIN was established in 2003 as a collaboration between the company owning Vision (In Practice Systems) and the CSD Medical Research Group (now Quintiles IMS). In April 2015, THIN reported covering 6% of the UK population, with 562 practices and 11 million total patients. There is a considerable overlap (around 60%) between CPRD and THIN practices, which has implications for studies wishing to replicate findings between different databases.[10] QResearch collects data from practices using EMIS systems and is the biggest PCD, with approximately 1500 practices in 2017, covering a population of more than 22 million patients.[11] ResearchOne is a collaboration between the provider of SystmOne, The Phoenix Partnership (TPP) and the University of Leeds, reporting 28 million (primary and secondary care) records and 423 practices in 2017.

The geographical coverage of PCDs is dependent on the location of practices using the parent clinical computer system, which is in turn dependent on historical patterns of market penetration by the software suppliers and system uptake by general practices. Geographical representativeness is an important prerequisite if analysts are to generalise PCD findings to the whole of England and the UK, which is what routinely happens in practice. This is due to great regional variability across England in terms of population characteristics (primarily: age, ethnicity and deprivation),[12] or even regional variation in hard outcomes. For example, a persistent mortality divide between North and the South of England has existed since the middle of the previous century,[13] while, more recently, much higher mortality rates were observed for young adults in the North of England.[14] There is also regional variation in the organisation and productivity of health services in England,[15 16] which could have important implications for the generalisability of health services research with the use of regionally unrepresentative PCDs. The aim of this paper is to describe the regional distribution of clinical computer systems in English primary care, evaluate the implications of the current picture of representativeness and provide some insight into the sustainability of existing PCDs.

## METHODS

### Data

Clinical computer system information was obtained from NHS Digital after direct communication, for August 2016. Primary care workforce and patient information as of 30 September 2016 was downloaded from the NHS Digital website.[17] At the practice level, information was available on geography (Clinical Commissioning Group (CCG) and NHS region), patient list size by age groups, and numbers and full-time equivalent for general practitioners (GPs) and nurses. Deprivation was quantified using the 2015 release of the Index of Multiple Deprivation (IMD), a complete aggregate measure widely used to quantify area deprivation, attributed to the practice location.[18] Spatial coordinates for NHS organisational units in 2016 were obtained from the Office for National Statistics open geography portal.[19] We focused on two organisational levels, the lower CCGs with 209 units, and the higher NHS regions with 14 units.

### Analyses

For all aspects of data manipulation and analysis we used Stata V.14.1. Whenever medians are reported, we also report the 25th and 75th centiles. Spatial maps were plotted using the *spmap* command.[20] Practice-level data were aggregated by clinical computer system, to provide information on all patients, patients aged ≥75 years, GPs and nurses, practice location deprivation and list size. Counts for each clinical computer system, by NHS region, were also calculated. Spatial graphs at the CCG level, with additional information on NHS regions, were plotted for the three most popular clinical computer systems, to provide a visual guide in regional distribution and representativeness.

## RESULTS

System information was missing for 49 (0.7%) of 7526 general practices. EMIS systems were used in 4199 practices (56%), with all but 23 of these using EMIS Web. SystmOne was used in 2552 (34%), Vision in 636 (9%) and Evolution in 90 (1%) practices. Patterns of area deprivation, based on the locations of general practices, were similar across all systems. SystmOne practices tended to

**Table 1** Regional distributions of systems and the characteristics of their respective general practices*†‡§

|  | EMIS¶ | SystmOne | Vision V.3 | Evolution |
|---|---|---|---|---|
| **Aggregates (%)** | | | | |
| Number of practices | 4199 (56) | 2552 (34) | 636 (9) | 90 (1) |
| Number of patients | 32 191 392 (56) | 20 199 414 (35) | 4 601 205 (8) | 6 29 166 (1) |
| Number of GPs | 18 675 (57) | 11 160 (34) | 2433 (7) | 393 (1) |
| **Medians (25th and 75th centiles)** | | | | |
| IMD 2015** | 22.2 (12.1, 37.4) | 22.5 (12.8, 36.8) | 22.4 (12.3, 37.0) | 22.7 (14.4, 31.0) |
| List size | 6833 (4257, 10 094) | 7080 (4214, 10 553) | 6279 (3988, 9759) | 6222 (4743, 9121) |
| Patients aged ≥75 years | 476 (240, 823) | 524 (256, 895) | 455 (225, 710) | 592 (400, 924) |
| **Means (SD)** | | | | |
| All GPs | 5.1 (3.4) | 5.1 (3.7) | 4.5 (3.1) | 5.5 (2.9) |
| Female GPs | 2.8 (2.4) | 2.7 (2.5) | 2.3 (2.1) | 2.7 (2.1) |
| GPs aged <40 years | 1.7 (1.9) | 1.6 (2.0) | 1.4 (1.7) | 1.5 (1.6) |
| GPs aged 40–54 years | 2.3 (1.9) | 2.4 (2.1) | 2.0 (1.8) | 2.8 (1.9) |
| GPs aged ≥55 years | 1.0 (1.0) | 1.0 (1.0) | 1.1 (1.0) | 1.2 (1.1) |
| All nurses | 3.1 (2.3) | 3.6 (2.6) | 2.8 (1.9) | 3.4 (1.6) |
| **Regional counts, NHS regions (%)** | | | | |
| Wessex | 164 (55) | 113 (38) | 17 (6) | 4 (1) |
| London | 917 (68) | 254 (19) | 182 (13) | 1 (0) |
| Yorkshire and the Humber | 186 (25) | 544 (74) | 5 (1) | 0 (0) |
| Cumbria and the North East | 270 (59) | 172 (38) | 12 (3) | 0 (0) |
| Cheshire and Merseyside | 353 (92) | 19 (5) | 8 (2) | 2 (1) |
| North Midlands | 260 (54) | 216 (45) | 2 (0) | 2 (0) |
| West Midlands | 496 (76) | 96 (15) | 58 (9) | 0 (0) |
| Central Midlands | 156 (28) | 378 (69) | 16 (3) | 0 (0) |
| East | 112 (21) | 413 (77) | 4 (1) | 4 (1) |
| South West | 225 (59) | 86 (22) | 7 (2) | 65 (17) |
| South East | 303 (56) | 96 (18) | 145 (27) | 1 (0) |
| South central | 227 (55) | 129 (31) | 57 (14) | 3 (1) |
| Greater Manchester | 310 (65) | 36 (8) | 123 (26) | 8 (2) |
| Lancashire | 220 (100) | 0 (0) | 0 (0) | 0 (0) |

*Data for August 2016 (clinical system) and September 2016 (GMS data).
†System information not available for 49 (0.65%) of 7526 practices.
‡All GP numbers exclude locums.
§SystmOne provided by TPP, Vision (version 3) provided by In Practice Systems, Evolution provided by Microtest.
¶EMIS includes Web (4176 practices), LV (19 practices) and PCS (4 practices).
**Index of Multiple Deprivation (higher score implies higher levels of deprivation); details available in the 2015 technical report of the English Indices of Deprivation.[18]
GMS, General Medical Services; GPs, general practitioners; IMD, Index of Multiple Deprivation; NHS, National Health Service.

be larger (median of 7080 patients), followed by EMIS (6833), Vision (6279) and Evolution (6222).

Great regional variability in system usage was observed both at the NHS region level (table 1) and CCG level (figures 1–3). EMIS is present in all but 18 of the 209 CCGs (91.4%), with a much stronger presence in the West of England, London and the South. SystmOne is present in 120 CCGs (57.4%), and is mainly active in the East and some regions in the South. Vision, although with a much lower market share than SystmOne, is still used in

96 CCGs (45.9%), mainly in London, the South, Greater Manchester and Birmingham. Evolution is only present in 18 CCGs (8.6%) and is primarily used in the South West.

## DISCUSSION

High regional variability exists in the use of different clinical computer systems in English primary care, which should be a consideration when utilising primary care

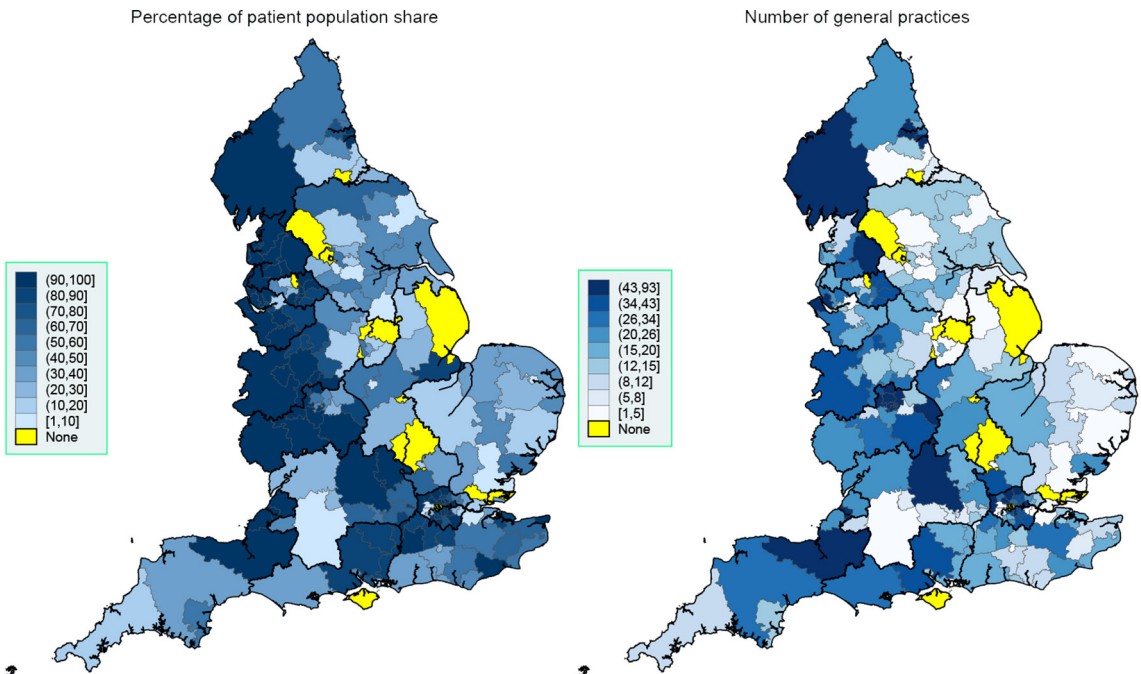

**Figure 1** Spatial map at the CCG level, September 2016: EMIS. Thicker border lines correspond to the 14 NHS regions, left graph uses equidistant class breaks; right graph uses class breaks based on distribution of variable of interest, with each class having approximately the same number of spatial polygons (CCGs). CCG, Clinical Commissioning Group; NHS, National Health Service.

electronic health databases based on this population in the future, especially if effect heterogeneity (or other forms of heterogeneity) is context relevant. For example, drawing nationwide conclusions in health services organisation would be more problematic than identifying

medication side effects. EMIS Web is by far the most widely used clinical computer system and therefore QResearch is the most nationally representative single database—potentially able to collect data from almost all English CCGs. SystmOne has a very strong presence in many parts

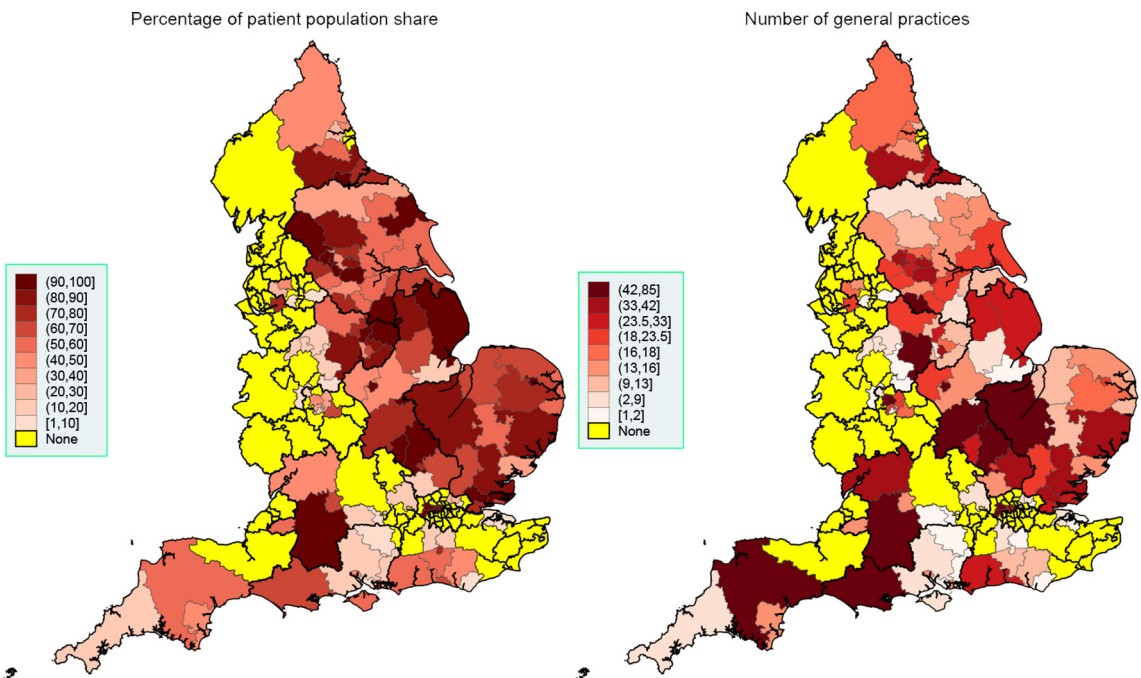

**Figure 2** Spatial map at the CCG level, September 2016: SystmOne. Thicker border lines correspond to the 14 NHS regions, left graph uses equidistant class breaks; right graph uses class breaks based on distribution of variable of interest, with each class having approximately the same number of spatial polygons (CCGs). CCG, Clinical Commissioning Group; NHS, National Health Service.

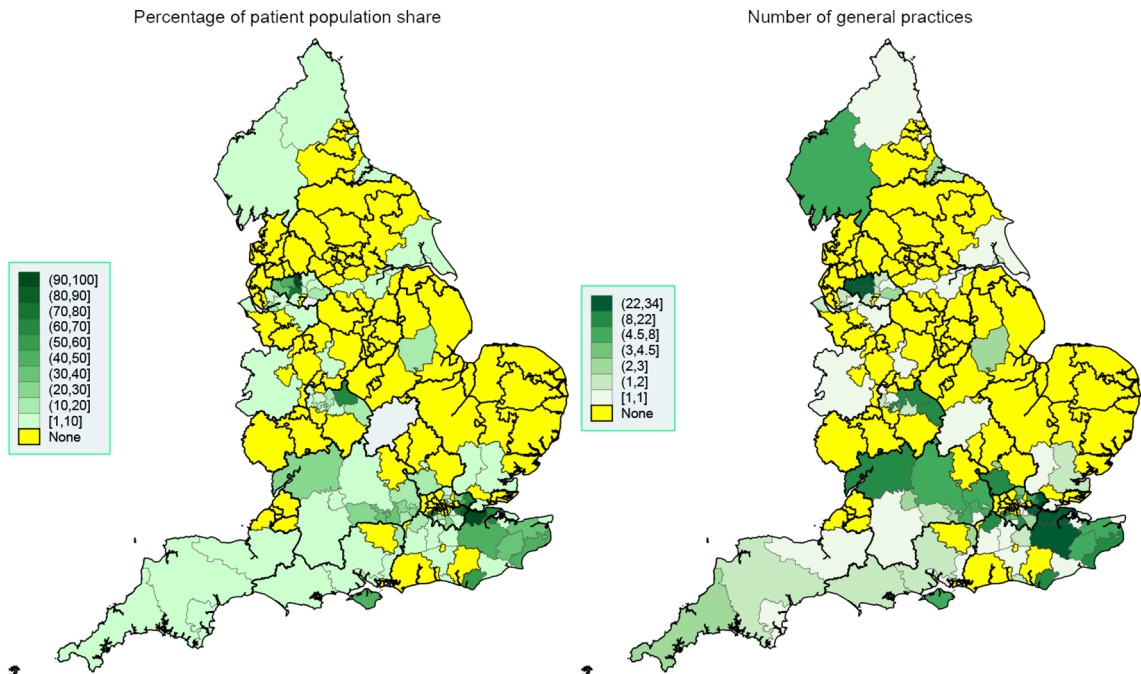

Percentage of patient population share

Number of general practices

**Figure 3** Spatial map at the CCG level, September 2016: Vision. Thicker border lines correspond to the 14 NHS regions, left graph uses equidistant class breaks; right graph uses class breaks based on distribution of variable of interest, with each class having approximately the same number of spatial polygons (CCGs). CCG, Clinical Commissioning Group; NHS, National Health Service.

of England, but no presence in many CCGs in the North West, West Midlands, London and South East. The ResearchOne database is therefore unable to capture data from many regions. Finally, Vision is the most geographically restricted of the three major clinical computer systems, with relatively few practices heavily concentrated in three conurbations and the South. The CPRD and THIN databases are therefore currently unable to provide comprehensive coverage of large parts of the country, particularly in the North and East of England.

### Strengths and limitations of the study

The main strength of this study is the use of numerous national administrative datasets of high data quality, allowing us to obtain a complete picture for the whole of England. The main weakness of the work is the fact we cannot equate the regional presence of a clinical system to active contribution to a primary care database—not all practices contribute data and contributing practices are anonymised—and we have therefore discussed potential contribution instead. Additional information on currently registered patients would have been relevant, but is not routinely available for non-users of the resources (but can be deduced by users).

### Findings and implications

The current picture of clinical system usage in English primary care is very different to what was reported for 2011.[2] Although EMIS is still the biggest provider and has retained its market share (56% in both 2011 and 2016), its LV and PCS systems which dominated the market in 2011 are hardly used anymore, with almost all practices

having transitioned to the Web system. The use of TPP's SystmOne has increased from 18% to 34%, while that of Vision by In Practice System has halved (from 18% to 9%). Many providers that were present in 2011 have subsequently withdrawn from primary care, with the exception of Microtest's Evolution (transitioned from Practice Manager). If the current trend continues, English primary care will be completely dominated by EMIS Web and SystmOne in the next 5–10 years, and access to both of these systems would ensure almost complete coverage for England.

The trend for primary care convergence to two clinical systems has implications for the future of PCDs and the research findings based on them. CPRD and THIN will need to adapt very quickly and include EMIS and/or SystmOne practices in their processes. Given that the CPRD and THIN are the two most widely used primary care databases in clinical research, losing them altogether, as happened with the DIN-LINK database,[21] would be a severe setback for the research community. As of 20 July 2017, a PubMed search identified 1782 published papers linked to the CPRD (886 in the last 5 years), 471 linked to THIN (303 in the last 5 years), 71 linked to QResearch (32 in the last 5 years) and two to ResearchOne (both in the last 5 years). Although not exhaustive, this search indicates the large variability across databases in terms of scientific contribution, demonstrating that the most accessible and productive databases are the ones at immediate risk.

Within the CPRD, there are clear actions towards future-proofing the resource, in light of the deterioration of the Vision market share. A large number of EMIS

practices are already contributing data to the resource, but differences in the data format (compared with the standard Vision format) has prevented their immediate release along with Vision data, while it was not possible to link the EMIS data to other data sets. Nevertheless, a major transformation in processes is being undertaken which will allow the release of both Vision and EMIS data as standard, within 2018. In addition, the recruitment of EMIS practices continues, with over 150 practices having joined the CPRD in the last 12 months.

Users of the UK PCDs need to be aware of the generalisability issues we described, and consider if there are any risks relevant to their studies. Generalisability (external validity) should be discussed as standard in such work and is listed as an item (no. 21) in both the Strengthening the Reporting of Observational Studies in Epidemiology and REporting of studies Conducted using Observational Routinely collected health Data statements.[22 23] The context is important here, and regional representativeness may be less relevant for clinical questions but more relevant for health services research. Sensitivity analyses on a more representative group of practices, obtained through deterministic sampling and existing software,[24] can also be used to strengthen findings.[25] However, the strong clustering of clinical systems within CCGs, largely driven top-down from CCGs to general practices, limits the usefulness of such sampling approaches.

## CONCLUSIONS

The geographical representativeness of primary care databases varies enormously, and the two most used databases in the UK, the CPRD and THIN, were in 2016 the least representative of the major databases due to the quickly diminishing market share of the clinical computer system providing their data (Vision). The existence of these databases is under threat, and urgent action is required to allow data collection from at least one of the two dominant clinical systems (EMIS Web and SystmOne). CPRD has recognised this, and has recently negotiated access to data held by EMIS practices, and is due to operationalise this data by 2018. In addition, development and access barriers that have restricted publication outputs from data drawn from EMIS (QResearch) and SystmOne (ResearchOne) practices urgently need to be overcome if the confidential use of NHS patient data is to continue driving research that directly informs patient safety, management and health services policy.

### Author affiliations
[1]NIHR School for Primary Care Research, University of Manchester, Manchester, UK
[2]Division of Population Health, Health Services Research and Primary Care, Faculty of Biology, Medicine and Health, University of Manchester, Manchester, UK
[3]Nuffield Department of Health Care Sciences, University of Oxford, Oxford, UK
[4]The Institute of Applied Health Sciences, University of Aberdeen, Aberdeen, UK
[5]Department of Public Health and Primary Care, School of Clinical Medicine, University of Cambridge, Cambridge, UK
[6]Department of Health Sciences, University of York, York, UK
[7]Division of Pharmacy and Optometry, Faculty of Biology, Medicine and Health, University of Manchester, Manchester, UK

**Acknowledgements** We thank the Office of National Statistics and NHS Digital for the wealth of information they have collected and systematically organised, which made this study possible.

**Contributors** EK designed the study, extracted the data from all sources, performed the analyses and drafted the first version of the manuscript. RJS, PJH, DE, TD and DMA critically edited the manuscript. EK is the guarantor of this work and, as such, had full access to all the data in the study and takes responsibility for the integrity of the data and the accuracy of the data analysis.

**Funding** MRC Health eResearch Centre Grant MR/K006665/1 supported the time and facilities of EK.

**Disclaimer** The views expressed are those of the authors and not necessarily those of the MRC.

**Competing interests** EK, RJS, PJH and DE are members of the Independent Scientific Advisory Committee (ISAC) for MHRA database research:

**Patient consent** Not required.

**Provenance and peer review** Not commissioned; externally peer reviewed.

**Data sharing statement** The data used in this study are freely available and the authors are happy to share an organised and cleaned final dataset.

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
