## [Reviewer comments · BMJ Open]

ARTICLE DETAILS

TITLE (PROVISIONAL)	Spatial distribution of clinical computer systems in primary care in England in 2016 and implications for primary care electronic medical record databases: a cross sectional population study
AUTHORS	Kontopantelis, Evangelos; Stevens, Richard; Helms, Peter; Edwards, Duncan; Doran, Tim; Ashcroft, Darren

VERSION 1 – REVIEW

REVIEWER	Dr Beverley Ellis University of Central Lancashire United Kingdom
REVIEW RETURNED	23-Nov-2017

GENERAL COMMENTS	The reviewer also provided an additional comments. Please contact the publisher for full details.
---

REVIEWER	Liam Smeeth LSHTM, UK
REVIEW RETURNED	24-Nov-2017

GENERAL COMMENTS	This is a well-written paper reporting carefully done work. I previously refereed it for the BJGP and the authors seem to have responded to some of my comments. While it is likely to be chiefly of interest to academic audiences, the information on the distribution and patterns of usage of the different software systems may well have broader interest. Specific comments. 1. Throughout, I would suggest it would be clearer to emphasise the size of databases in terms of currently registered patients, rather than the totality of historical records. The period over which data are available and the total number of records can perhaps be given once, but as secondary information. 2. Table 1 is sometimes hard to follow. For example, the patients aged over 75 row: is this the average number per practice? Similarly, the All GPs row: is this the mean doctors per practice? Liam Smeeth
--

VERSION 1 – AUTHOR RESPONSE

Reviewer: 1

Reviewer Name: Dr Beverley Ellis

1. You may be interested in attached paper Further study could consider PCD validation and quality indicators.

Response: Thank you for bringing this to our attention. This paper is an older piece of work that highlighted the importance of high quality EHRs in the NHS and described what had been published at the time (2003) using EHRs, through a systematic review. We are now using this reference in the introduction where we describe the diverse role EHRs play in medical research.

Reviewer: 2

Reviewer Name: Liam Smeeth

This is a well-written paper reporting carefully done work. I previously refereed it for the BJGP and the authors seem to have responded to some of my comments. While it is likely to be chiefly of interest to academic audiences, the information on the distribution and patterns of usage of the different software systems may well have broader interest.

Specific comments.

1. Throughout, I would suggest it would be clearer to emphasise the size of databases in terms of currently registered patients, rather than the totality of historical records. The period over which data are available and the total number of records can perhaps be given once, but as secondary information.

Response: The reviewer is quite right but unfortunately that information does not appear to be made available as standard by the database owners (obviously, users can deduce it). Searching the relevant websites were unsuccessful and we think the information may be considered commercially sensitive. We have expanded the limitations section to discuss this.

2. Table 1 is sometimes hard to follow. For example, the patients aged over 75 row: is this the average number per practice? Similarly, the All GPs row: is this the mean doctors per practice?

Response: Thank you, we have made edits to the table to clarify these issues. Each section is preceded by a row where that information is provided (% , mean and SD or median and IQR). These are now in bold. We are happy to address any other issues regarding the table at the production stage.

VERSION 2 – REVIEW

REVIEWER	Dr Beverley Ellis University of Central Lancashire UK
REVIEW RETURNED	04-Dec-2017

GENERAL COMMENTS	The author has addressed minor revision required following reviewer comments, to the best of my knowledge.
--

REVIEWER	Liam Smeeth LSHTM UK
REVIEW RETURNED	02-Dec-2017

GENERAL COMMENTS	The authors have dealt with my concerns.
--